# InDel and CNV within the *AKAP13* Gene Revealing Strong Associations with Growth Traits in Goat

**DOI:** 10.3390/ani13172746

**Published:** 2023-08-29

**Authors:** Xiaoyue Song, Yangyang Bai, Rongrong Yuan, Haijing Zhu, Xianyong Lan, Lei Qu

**Affiliations:** 1Shaanxi Provincial Engineering and Technology Research Center of Cashmere Goats, Yulin University, Yulin 719000, China; songxiaoyue@yulinu.edu.cn (X.S.); bai345@126.com (Y.B.); rongrong454250833@163.com (R.Y.); haijingzhu@yulinu.edu.cn (H.Z.); 2College of Life Sciences, Yulin University, Yulin 719000, China; 3College of Animal Science and Technology, Northwest A&F University, Xianyang 712100, China

**Keywords:** goat, *AKAP13*, InDel, CNV (copy number variation), growth trait

## Abstract

**Simple Summary:**

The *AKAP13* gene has been found to be related to bone formation. As an important gene regulating growth and development, whether *AKAP13* has any influence on the growth traits of goats is still unknown. Therefore, we hope to explore the InDel and CNV genetic variation of the *AKAP13* gene in Shaanbei white cashmere goats and their relationship with growth traits to find effective molecular marker sites.

**Abstract:**

A-kinase-anchoring protein 13 (*AKAP13*) is a member of the AKAP protein family that has been found to be associated with bone formation. Thus, we investigated the *AKAP13* gene as a potential candidate gene for molecular-marker-assisted selection (MAS) in breeding. Our aim was to explore genetic variations (InDel and CNV) within the *AKAP13* gene of Shaanbei white cashmere (SBWC) goats and analyze their relationship with growth traits. Ultimately, we identified three InDel loci (16-bp deletion, 15-bp insertion, and 25-bp deletion) and three CNVs, and the 16-bp and 15-bp loci were significantly associated with goat body length (*p* < 0.05). Both the 16-bp deletion variant and the 15-bp insertion variant facilitated an increase in body length in goats. In addition to this, there was a certain superposition effect between 16-bp and 15-bp loci, although there was no linkage. Additionally, the CNV1 locus was significantly correlated with body height and body length of goats (*p* < 0.05), and CNV2 was significantly correlated with chest depth, chest circumference, and cannon circumference of goats (*p* < 0.05). Individuals with gain type showed excellent growth performance. In conclusion, the InDel and CNV loci that we have identified could possibly serve as effective molecular markers in goat breeding, which is very essential for improving efficiency and success of breeding. Moreover, our findings provide a new avenue for further research into the function of the *AKAP13* gene.

## 1. Introduction

With the increase in the world population and economic development, the demand for livestock and poultry products has significantly increased [1,2]. However, it is difficult to provide an adequate supply of quality goat products due to various factors, such as traditional breeding methods and inefficient production constraints [3,4,5,6]. Therefore, in addition to improving feeding methods and management conditions, selecting goat breeds with excellent growth traits is an effective means of promoting the development of the goat industry and improving economic benefits. Shaanbei white cashmere (SBWC) goats are an important and exceptional breed known for their meat and fleece production, and they have been selectively bred in China. SBWC goats are recognized for their rapid growth, resilience to rough feeding conditions, cold tolerance, adaptability, and disease resistance [7]. With advancements in molecular biology technology, molecular-marker-assisted selection (MAS) breeding has gained momentum and has been applied in industry. Using molecular markers like insertion/deletion (InDel) and copy number variation (CNV) has shown great advantages over traditional breeding [8,9,10]. Numerous studies have demonstrated the effective use of InDel and CNV markers in association with growth traits, thereby reducing breeding time and improving outcomes [11].

The rapid progress of high-throughput sequencing technology, along with the continuous improvement of public databases and websites, has made it feasible to identify candidate genes associated with production traits [12,13,14]. Herein, A-kinase-anchoring proteins (AKAPs) were chosen because they are a family of proteins with the ability to regulate signal transduction processes [15,16]. AKAPs have been found to be widely involved in the regulation of growth, development, metabolism, and reproduction [17,18,19]. The main reason for their versatility is that they contain the PKA-binding region and other signaling molecule binding domains, which can bind a variety of signaling molecules, including kinases, phosphatases, adenylate cyclases, GTPases, membrane receptors, and other regulatory proteins, forming a multisignaling complex and regulating their activities [20]. *AKAP13*, as a significant member of this protein family, exhibits similar characteristics. Besides its interaction with PKA, *AKAP13* can interact with various protein factors, such as MEK/ERK, p38, and RhoA [21,22]. This suggests that *AKAP13* is capable of regulating various life processes, including growth. It has been shown that the *AKAP13* gene affects heart development in mice, and bone formation and *AKAP13* deficiency can lead to osteoporosis and abnormal heart development [23,24]. Moreover, the absence of the *AKAP13* gene has been linked to weight gain in mice [25]. Hence, the *AKAP13* gene is a potential candidate associated with growth traits.

While previous studies have verified that *AKAP13* may be associated with growth, there is a lack of research on the effect of the *AKAP13* gene InDel and CNV variants on growth. Therefore, the objective of this study is to investigate the InDel and CNV loci within the *AKAP13* gene and systematically analyze their relationship with growth traits in goats. We aim to identify new molecular markers for MAS breeding in goats and provide foundational data for further understanding the function of the *AKAP13* gene.

## 2. Materials and Methods

All animal tests performed in this study were conducted under the supervision and guidance of the Faculty of Animal Policy and Welfare Committee of Northwest A&F University, and all procedures were in accordance with the specifications (NWAFU-314020038).

### 2.1. Animal Samples Collection

SBWC goats were kept in similar and suitable conditions at farms in Yulin, Shaanxi, China. A total of 487 ear samples were collected from the adult female goats (2–3 years old) and immediately preserved in liquid nitrogen until used. Various growth traits, including body length, body height, height at hip cross, chest circumference, chest depth, chest width, cannon circumference, and hip width, were measured and recorded for each goat. 

### 2.2. Genomic DNA Isolation

Genomic DNA was extracted from the ear tissue samples of the goats using the modified salt–chloroform extraction method [26,27]. The concentration and purity of the DNA samples were assessed using Nanodrop 2000 (Thermo Scientific, Waltham, MA, USA). DNA of acceptable quality was used for subsequent experiments. The DNA samples were then diluted to 20 ng/µL with ddH_2_O. In total, 30 samples were randomly selected to construct a DNA mixing pool for preliminary detection of polymorphism at screened InDel sites [28,29].

### 2.3. Primer Design and Genotype Detection

First, we searched the Ensembl database’s Variant table (https://asia.ensembl.org/ accessed on 4 March 2022) for information on eight InDel loci for the *AKAP13* gene in goats. Then, according to the reference sequence of the goat *AKAP13* gene (GenBank No: NC_030828.1), eight pairs of primers were designed using NCBI Primer-BLAST (https://www.ncbi.nlm.nih.gov/tools/primer-blast/ accessed on 8 March 2022) and synthesized by Sangon Biotech, China (Table 1). Three CNV loci selected from the Animal Omics Database (http://animal.nwsuaf.edu.cn/ accessed on 10 March 2022) were also explored—namely, CNV1, CNV2, and CNV3 with the *MC1R* gene as the reference gene (Table 2 and Table 3).

### 2.4. InDel and CNV Genotyping of AKAP13 Gene

Referring to the previous study, the touch-down PCR procedure was used for amplification. After the end, 6.0 µL PCR products were electrophoresed on a 3.0% agarose gel, which was stained with nucleic acid dyes (Tsingke, Beijing, China) and examined for each individual.

Three controls were set for each sample, and then the quantitative real-time polymerase chain reaction (qPCR) results of the *AKAP13* gene copy number were analyzed. The qPCR amplification reaction system volume and procedure were as described previously [30], and the result was processed using the method 2 × 2^−ΔCt^; 2 × 2^−ΔCt^ > 2 was gain type, it < 2 was loss type, and it = 2 was median type. Additionally, the gene structure of the goat *AKAP13* gene was drawn as shown in Figure 1.

### 2.5. Statistical Analyses

The SHEsis program was used to analyze whether these InDel loci were in Hardy–Weinberg equilibrium (HWE). To calculate the genetic parameters, including Ho (observed heterozygosity), He (expected heterozygosity), Ne (effective allele numbers), and PIC (polymorphism information content), the Nei method was used directly [31]. Simplified linear models were used to determine the relationship between goat genotypes and each growth trait:Y_ij_ = μ + G_i_ + E_ij_
where Y_ij_ represented the measurement of growth traits for each goat, μ represented the overall mean, G_i_ represented the effect of genotype, and E_ij_ represented the random error. One-way ANOVA was used to analyze the differences in growth traits between different genotypes and combined genotypes of goats using SPSS software (version 24.0, IBM Corp, Armonk, NY, USA).

### 2.6. Genetic Linkage Analysis

Linkage disequilibrium (LD) analysis was performed on InDel loci using the SHEsis platform [32]. D′ and r^2^ were employed to assess the degree of linkage between InDel loci. 

## 3. Results

### 3.1. Characterization of Three InDel and CNV Loci in the AKAP13 Gene of SBWC Goats

In this study, three InDel loci were identified. The P1 locus was a 16-bp deletion (NC_030828.1:g.15695906-15695921delAGTGATAGTCGGAGGA), and three genotypes were detected: II (insertion/insertion, 212/212 bp), ID (insertion/deletion, 212/196 bp), and DD (deletion/deletion, 196/196 bp) (Figure 2A). While the P2 InDel locus was a 15-bp insertion (NC_030828.1:g.15811839-15811840 insCCCAGGGCCTCACAC) and also showed three genotypes: including II (insertion/insertion, 240/240 bp), ID (insertion/deletion, 225/240 bp) and DD (deletion/deletion, 225/225 bp) (Figure 3A). P3 InDel locus was a 25-bp deletion (NC_030828.1:g.15544608-15544632delGATCTTTCCTTTCCTTTCCTTTGTT). There were three genotypes: including II (insertion/insertion, 243/243 bp), ID (insertion/deletion, 218/243 bp), and DD (deletion/deletion, 218/218 bp) (Figure 4A). DNA sequencing maps revealed mutant site sequences (Figure 2B, Figure 3B and Figure 4B).

There are three CNVs in the *AKAP13* gene, which are named CNV1 (NC_030828.1:g. 15,686,001–15,687,600, 1599 bp), CNV2 (NC_030828.1:g.15,510,001–15,512,000, 1999 bp), and CNV3 (NC_030828.1:g. 15,580,401–15,582,400, 1999 bp), respectively. The CNV primers were detected via qPCR, and the melting curve showed that the quantitative primers belonged to specific amplification and had good amplification efficiency (Figure 5).

### 3.2. InDel Detection: Genotype Frequency, Linkage Disequilibrium, and Haplotype Analyses of the Goat AKAP13 Gene

All genetic parameters had been calculated and presented in (Table 4). For the P1 InDel locus, the frequencies of the allele “I” and allele “D” were 0.884 and 0.116, respectively. The genotype distribution was not in HWE (*p* < 0.05). For the P2 InDel locus, DD genotype was much more frequent than II and ID genotypes, and the genotype distribution was in accordance with HWE (*p* > 0.05). The PIC values indicated that both InDel loci were at low degree of polymorphism (0 < PIC< 0.25). For the P3 InDel locus, it showed medium genetic diversity (0.25 < PIC < 0.5), and “D” allele frequency was higher than “I” allele frequency. The genotype distribution was in accordance with HWE (*p* > 0.05).

The results of the LD analysis are shown in (Figure 6) the D′ value and r^2^ value were calculated (Table 5). These results demonstrated that there was no LD in P1 and P2 InDel loci in this study. Seven haplotypes were detected using haplotype analysis, including haplotype1 (I_P1_D_P2_I_P3_), haplotype2 (I_P1_D_P2_D_P3_), haplotype3 (I_P1_I_P2_I_P3_), haplotype4 (I_P1_I_P2_D_P3_), haplotype5 (D_P1_D_P2_D_P2_), haplotype6 (D_P1_I_P2_I_P2_), and haplotype7 (D_P1_I_P2_D_P2_) (Table 6). The haplotype D_P1_D_P2_I_P3_ was not found. Following this, we analyzed the association of nine combined genotypes with growth traits in SBWC goats (Table 7). The body length of goats with combined genotype D_P1_D_P1_-I_P2_D_P2_ was significantly greater than other combined genotypes (*p* < 0.05). This also demonstrated that these two InDel loci can significantly promote the growth of SBWC goats.

### 3.3. CNV Detection: Frequency of the Goat AKAP13 Gene Genotypes

According to the 2 × 2^−∆CT^ method, we divided the CNV types into three classes, including Loss type (0~2), Medium type (2) and Gain type (>2). There are three types (gain, loss, and medium) of CNV1 and CNV2, but only one type of CNV3 (gain) (Table 8). In addition, among the three CNVs, gain type frequency was the highest and was 0.886, 0.692, and 1, respectively.

### 3.4. Association Analysis of Mutations with Growth Traits

The results of the one-way ANOVA showed that only P1 and P2 were significantly associated with body length in SBWC goats (Table 9). For the P1 InDel locus, goats with ID and DD genotypes had significantly higher body lengths than those with II genotype (*p* = 0.01). This implies that 16-bp deletion had a beneficial effect on goat growth traits. However, for the InDel locus at P2, the body length of goats with the ID genotype was significantly higher than those with the DD genotype (*p* = 0.014). This means that this 15-bp insertion was a favorable mutation. The superiority of heterozygous genotypes in body length may be due to the complementary effect caused by the comprehensive aggregation of dominant genes in the hybrid offspring.

The statistical analyses showed that CNV1 was significantly associated with body height (*p* = 0.045) and body length (*p* = 0.046) of goats. For CNV2 locus, there was an association with chest depth (*p* = 0.011), chest circumference (*p* = 0.026), and cannon circumference (*p* = 0.014). For CNV1 and CNV2, the gain type is the higher-frequency type, which had better growth performance than the loss and medium types (Table 10).

## 4. Discussion

In this study, we found for the first time that two InDel loci in the *AKAP13* gene were significantly associated with growth traits in SBWC goats. Our previous studies have demonstrated that InDel variants in the *AKAP13* gene can affect litter size in goats [33]. Interestingly, this study also supported our conjecture that the *AKAP13* gene could serve as a valid molecular marker for association with growth traits. The three InDel loci we identified belonged to different types, P1 and P3 were deletion mutations, and P2 was an insertion mutation. There was no LD between these three loci. Moreover, based on the PIC values (0 < PIC < 0.25), P1 and P2 were polymorphic to a low degree, indicating low heterozygosity and possibly strong selection [34]. The P2 and P3 were in accordance with HWE, while P1 was not. Thus, this suggested that the P1 locus was subjected to a significant degree of environmental or human selection several generations ago, which contributed to the mutations and caused these effects [35].

The results of one-way ANOVA showed that P1 and P2 were significantly associated with goat body length. For the P1 locus, goats with ID and DD genotypes had significantly higher body lengths than the II genotype (*p* < 0.05), suggesting that the presence of the “D” allele significantly promoted the growth of goats. In addition, we found that this mutation may have a dosage effect. For the P2 locus, the body length of goats with the ID genotype was significantly higher than that of the DD genotype (*p* < 0.05). While the II genotype was larger than the DD genotype, it did not reach significance (*p* > 0.05). Therefore, we speculated that the effect of the heterozygous advantage leaded to such a result. The results of the combined genotype analysis showed that goats with the D_P1_D_P1_–I_P2_D_P2_ genotype exhibited the longest body length. This suggested that although P1 and P2 were not linked, there may be a functional complementarity of these two mutations. 

We used the CNVcaller software (https://github.com/JiangYuLab/CNVcaller, accessed on 25 May 2022) to analyze existing whole genome resequencing results and found these three CNVs in the world’s goat breeds [36,37], with significant copy number variations in Nubian goat, Longwood goat, and one cashmere goat (Appendix A). The resequencing results verified the presence of two loci in Shaanbei white Cashmere goats, after which the population was expanded. The results showed three types (gain, loss, and medium) of CNV1 and CNV2 but only one type of CNV3 (gain). Furthermore, the CNV1 was significantly associated with body height (*p* = 0.045) and body length (*p* = 0.046) of goats. For the CNV2 locus, there was an association with chest depth (*p* = 0.011), chest circumference (*p* = 0.026), and cannon circumference (*p* = 0.014). For CNV1 and CNV2, the gain type is the more frequent type, which has better growth performance than the loss and medium types. Thus, this result further validated our previous findings.

*AKAP13* is a 10 kDa protein with a PKA-anchoring domain at its N-terminal end, a guanine nucleotide exchange factor (GEF), and a nuclear receptor interaction domain (NRID) at its C-terminal end [38,39]. Previous studies have found that *AKAP13* is highly expressed in skeletal muscle and bone, suggesting that *AKAP13* may play an important role in the regulation of skeletal development [16]. Bone formation is a rigorous and complex process throughout animal life, including the regulation of osteoblasts and osteoclasts. Activation of the cAMP/PKA/CREB signaling pathway promotes differentiation of MSCs to osteoblasts, thereby promoting bone growth [40]. Thus, *AKAP13* can further influence the bone differentiation process by affecting PKA. It is well known that osteoblast differentiation is regulated by various hormones and cytokines, and Rho is one of the most important protein families [25]. It is worth noting that recent studies illustrated that a complete RhoA signaling process is required for bone formation [41]. *AKAP13* can activate RhoA through the GEF structural domain and thus have an important contribution to the differentiation process of osteoblasts [42]. Therefore, we hypothesized that variants in the *AKAP13* gene may affect the activity of the RhoAGEF structural domain, which in turn affects the activity of RhoA and osteoblast differentiation. On the other hand, runt-related transcription factor 2 (Runx2) is a major regulatory and transcriptional factor for bone development, and the InDel within the *Runx2* gene has been found to be associated with reproduction and growth in goats [7,40]. Earlier studies have shown that deletion of the *AKAP13* gene significantly suppresses its expression [43]. Therefore, mutations in the *AKAP13* gene may affect bone development by affecting the expression of *Runx2*. In addition to this, estrogen is also very important for the bone formation process, and lack of estrogen can lead to decreased bone mass and osteoporosis [44]. *AKAP13* can significantly enhance the ligand-dependent activity of estrogen receptors (ERs) [16]. Thus, variation within the *AKAP13* gene may affect the binding of *AKAP13* to ERs, affecting ER activity and interfering with the function of estrogen. Therefore, mutations in the *AKAP13* gene affecting bone growth by affecting the action of estrogen are also one of the possible pathways.

Although both InDel loci we identified were located in the intron region, they can also affect the alteration of animal traits. Mutation in the intronic region might affect the binding of the DNA sequence and DNA binding factors [45,46] as well as the transcriptional efficiency and the stability of mRNA [47,48]. Liu et al. (2020) found that the InDel locus located in the intron of the prolactin receptor (*PRLR*) gene is associated with growth in goats [49]. Wang et al. (2020) also found that variants located in the intron region of the *IGF2BP1* gene were associated with growth traits in goats [50]. Thus, our study further corroborated this conjecture.

## 5. Conclusions

In conclusion, in this study, we identified three InDel loci and three CNVs of the *AKAP13* gene in SBWC goats and found that the four InDel and CNVs were significantly associated with growth traits and could possibly serve as effective molecular markers for future MAS breeding.

## Figures and Tables

**Figure 1 animals-13-02746-f001:**
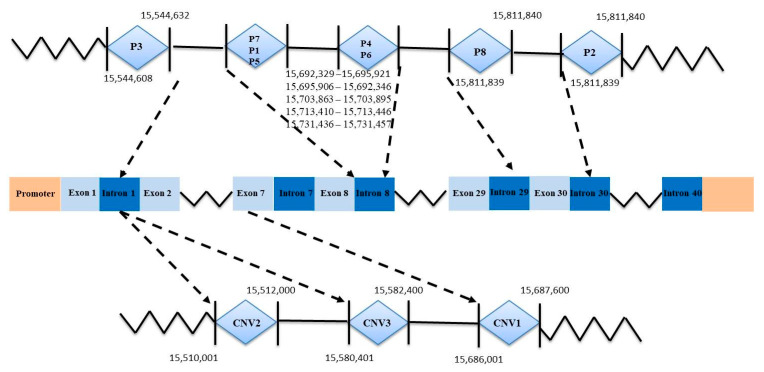
Gene structure of the goat *AKAP13* gene.

**Figure 2 animals-13-02746-f002:**
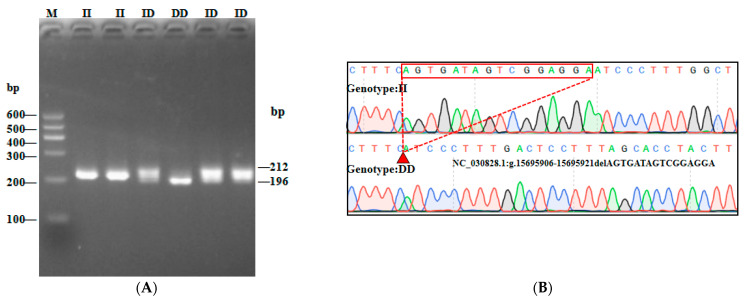
Agarose gel electrophoresis (**A**) and DNA sequencing map (**B**) of goat *AKAP13* gene locus P1-16 bp. Note: M, DNA marker.

**Figure 3 animals-13-02746-f003:**
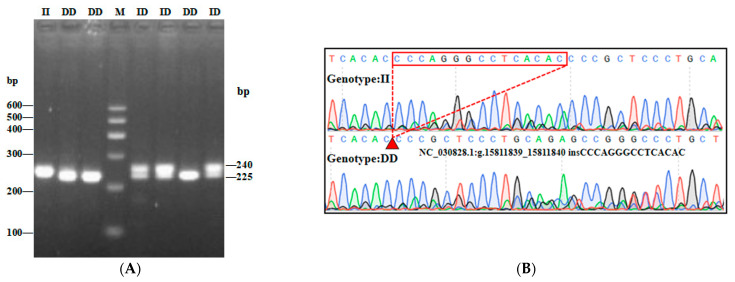
Agarose gel electrophoresis (**A**) and DNA sequencing map (**B**) of the goat *AKAP13* gene locus P2-15 bp. Note: M, DNA marker.

**Figure 4 animals-13-02746-f004:**
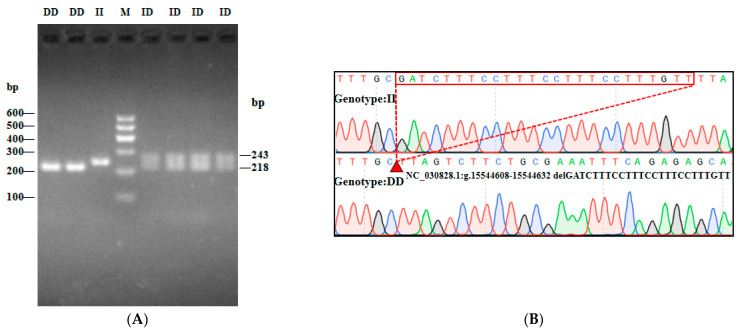
Agarose gel electrophoresis (**A**) and DNA sequencing map (**B**) of the goat *AKAP13* gene locus P3-25 bp. Note: M, DNA marker.

**Figure 5 animals-13-02746-f005:**
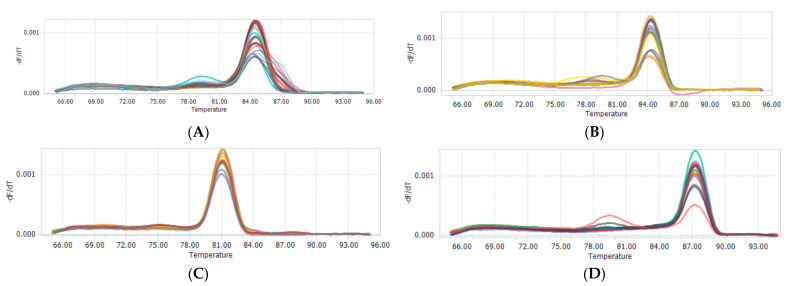
Melting curve of CNV1 (**A**), CNV2 (**B**), and CNV3 (**C**) sites of the *AKAP13* and MC1R (**D**) genes.

**Figure 6 animals-13-02746-f006:**
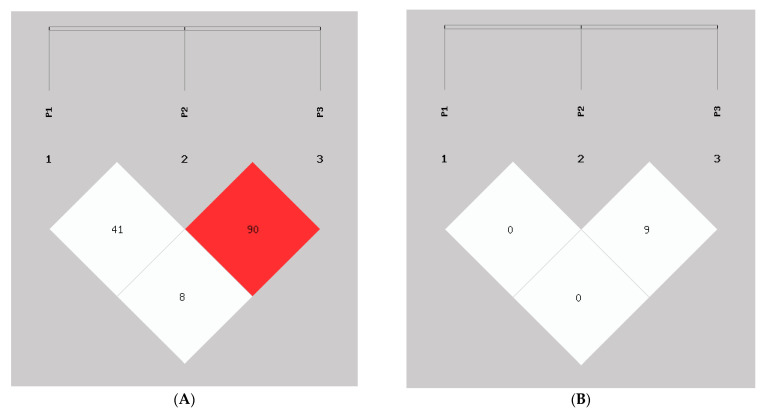
Linkage disequilibrium (LD) plot of the *AKAP13* gene three InDel loci in goat, P1, P2, and P3. (**A**) D′ value, (**B**) r^2^ value.

**Table 1 animals-13-02746-t001:** The primer information of PCR amplification of the *AKAP13* gene.

Loci	Primer Sequence (5′–3′)	Region	Size (bp)	Tm (°C)
P1-16 bp	F: ACAGCCCTGAATGATGGATACAC	Intron	196/212	TD-PCR
	R: TTCAGCAAAAGCAAACTCTCTGG			
P2-15 bp	F: GTTCAGGGCAGAGTGTGCTT	Intron	225/240	TD-PCR
	R: CACCCAGTAGCACCAAAGGG			
P3-25 bp	F: TGGAATTGGGATGTTTTGTGTG	Intron	218/243	TD-PCR
	R: AGCCACAGATTCCGAGCTTA			
P4-37 bp	F: CAGAGATAACCAGAGGAGGTGG	Intron	159/196	TD-PCR
	R: ACGCTGGGAAAAAGTCAGGT			
P5-33 bp	F: TCTGGTTTGGGTGCCATACT	Intron	175/208	TD-PCR
	R: TCATGTTGAAAGGGCATGCATTA			
P6-22 bp	F: GTGTCTTGTAATAACCTATAATGGGA	Intron	201/223	TD-PCR
	R: GCTGCTCTTACCTGTTTTGATG			
P7-18 bp	F: GTATTATTTCAAAGTTCATCCATG	Intron	214/232	TD-PCR
	R: TATAGAATTACCACATGAGCCA			
P8-15 bp	F: CCCCCAGCTGAGGAATTGAG	Intron	138/153	TD-PCR
	R: GAGGATGAACAACAGGGAGACA			

Note: TD-PCR, touch-down polymerase chain reaction.

**Table 2 animals-13-02746-t002:** The primers used for detection of the CNV mutations and the relative expression of the *AKAP13* gene.

Primers	Sequences (5′–3′)	Sizes (bp)
CNV1	F: CTTGAGCAGTGCTTTGCTGG	110
	R: CCCTCAAACGGTTGCTTGTG	
CNV2	F: TTTCCAGCCTGTTGACTCCG	117
	R: ACCAAGCCACTTCACCCAAT	
CNV3	F: ACGAACCTCTGCTTTCAACC	120
	R: TAGGATCGAAGGTGTCCTGG	
MC1R	F: GGCCTGAGAGGGGAATCACA	126
	R: AGTGGGTCTCTGGATGGAGG	

**Table 3 animals-13-02746-t003:** Information on the copy number variations within the goat *AKAP13* gene.

Primers	Chromosome	Start	End	Length	Location
CNV1	21	15,686,001	15,687,600	1599	exonic
CNV2	21	15,510,001	15,512,000	1999	intron
CNV3	21	15,580,401	15,582,400	1999	intron

**Table 4 animals-13-02746-t004:** Genetic parameters for three InDel loci of the *AKAP13* gene.

Loci	Frequencies				Ho	He	Ne	PIC	HWE *p*-Value
	Genotypes		Alleles						
P1-16 bp (*n* = 408)	II	0.802 (*n* = 327)	I	0.884	0.164	0.206	1.259	0.185	0.00025
	ID	0.164 (*n* = 67)	D	0.116					
	DD	0.034 (*n* = 14)							
P2-15 bp (*n* = 400)	II	0.025 (*n* = 10)	I	0.169	0.288	0.281	1.390	0.241	0.8844
	ID	0.2875 (*n* = 115)	D	0.831					
	DD	0.6875 (*n* = 275)							
P3-25 bp (*n* = 235)	II	0.089 (*n* = 21)	I	0.347	0.515	0.453	1.828	0.350	0.1121
	ID	0.515 (*n* = 121)	D	0.653					
	DD	0.396 (*n* = 93)							

Note: Ho, observed heterozygosity; He, expected heterozygosity; Ne, effective allele numbers; PIC, polymorphism information content; HWE, Hardy–Weinberg equilibrium; II, insertion/insertion; ID, insertion/deletion; DD, deletion/deletion.

**Table 5 animals-13-02746-t005:** D′ and r^2^ value of LD of the *AKAP13* gene in SBWC goats.

Loci	D′		r^2^	
	P2	P3	P2	P3
P1	0.417	0.084	0.009	0.003
P2		0.904		0.096

**Table 6 animals-13-02746-t006:** Haplotypic frequencies within the *AKAP13* gene in SBWC goats.

Haplotypic Names	Haplotypic Types	Haplotypic Frequencies
Haplotype1	I_P1_D_P2_I_P3_	0.004
Haplotype2	I_P1_D_P2_D_P3_	0.014
Haplotype3	I_P1_I_P2_I_P3_	0.069
Haplotype4	I_P1_I_P2_D_P3_	0.105
Haplotype5	D_P1_D_P2_D_P2_	0.161
Haplotype6	D_P1_I_P2_I_P2_	0.277
Haplotype7	D_P1_I_P2_D_P2_	0.369

**Table 7 animals-13-02746-t007:** The association of goat *AKAP13* P1-16 bp and P2-15 bp InDel loci combined genotypes and body growth traits of SBWC goats.

Body Measurement Traits	Combined Genotypes (Mean ± SE)/(Frequencies)								
	I_P1_I_P1_–I_P2_I_P2_ (0.139)	I_P1_I_P1_–I_P2_D_P2_ (0.182)	I_P1_I_P1_–D_P2_D_P2_ (0.248)	I_P1_D_P1_–I_P2_I_P2_ (0.032)	I_P1_D_P1_–I_P2_D_P2_ (0.076)	I_P1_D_P1_–D_P2_D_P2_ (0.141)	D_P1_D_P1_–I_P2_I_P2_ (0.010)	D_P1_D_P1_–I_P2_D_P2_ (0.053)	D_P1_D_P1_–D_P2_D_P2_ (0.119)
Body height (cm)	56.31 ± 0.25 (*n* = 332)	56.48 ± 0.20 (*n* = 435)	56.24 ± 0.18 (*n* = 595)	56.68 ± 0.43 (*n* = 77)	56.88 ± 0.27 (*n* = 180)	56.26 ± 0.23 (*n* = 340)	56.69 ± 0.85 (*n* = 24)	56.97 ± 0.32 (*n* = 127)	56.19 ± 0.26 (*n* = 287)
Body length (cm)	65.07 ± 0.38 ^bce^ (*n* = 332)	65.76 ± 0.33 ^bce^ (*n* = 415)	65.55 ± 0.28 ^bce^ (*n* = 574)	67.02 ± 0.66 ^ae^ (*n* = 75)	67.80 ± 0.42 ^ad^ (*n* = 158)	66.40 ± 0.33 ^bcde^ (*n* = 317)	68.34 ± 1.50 ^ac^ (*n* = 22)	68.48 ± 0.52 ^a^ (*n* = 105)	66.38 ± 0.38 ^bcde^ (*n* = 264)
Height at hip cross (cm)	59.95 ± 0.33 (*n* = 164)	60.15 ± 0.26 (*n* = 235)	59.85 ± 0.23 (*n* = 319)	60.43 ± 0.49 (*n* = 56)	60.54 ± 0.31 (*n* = 127)	59.92 ± 0.27 (*n* = 211)	61.20 ± 0.92 (*n* = 15)	60.72 ± 0.36 (*n* = 86)	59.87 ± 0.30 (*n* = 170)
Chest circumference (cm)	90.08 ± 2.03 (*n* = 334)	91.61 ± 2.14 (*n* = 438)	90.30 ± 1.16 (*n* = 596)	90.31 ± 0.94 (*n* = 77)	93.88 ± 3.60 (*n* = 181)	90.52 ± 0.46 (*n* = 339)	89.33 ± 1.74 (*n* = 24)	95.17 ± 5.08 (*n* = 128)	90.48 ± 0.51 (*n* = 286)
Chest depth (cm)	27.07 ± 0.16 (*n* = 331)	27.18 ± 0.14 (*n* = 427)	27.14 ± 0.12 (*n* = 537)	26.94 ± 0.44 (*n* = 74)	27.28 ± 0.27 (*n* = 170)	27.16 ± 0.18 (*n* = 280)	28.45 ± 0.46 (*n* = 21)	27.71 ± 0.29 (*n* = 117)	27.36 ± 0.17 (*n* = 227)
Chest width (cm)	17.48 ± 0.14 (*n* = 331)	17.58 ± 0.13 (*n* = 427)	17.55 ± 0.11 (*n* = 538)	18.11 ± 0.24 (*n* = 74)	17.99 ± 0.18 (*n* = 170)	17.77 ± 0.14 (*n* = 281)	18.31 ± 0.40 (*n* = 21)	17.97 ± 0.22 (*n* = 117)	17.71 ± 0.16 (*n* = 228)
Cannon circumference (cm)	8.02 ± 0.05 (*n* = 334)	8.08 ± 0.05 (*n* = 438)	8.10 ± 0.04 (*n* = 596)	8.23 ± 0.09 (*n* = 77)	8.25 ± 0.06 (*n* = 181)	8.21 ± 0.04 (*n* = 339)	8.17 ± 0.15 (*n* = 24)	8.24 ± 0.07 (*n* = 128)	8.21 ± 0.05 (*n* = 286)
Hip width (cm)	20.73 ± 0.18 (*n* = 159)	20.80 ± 0.14 (*n* = 229)	20.70 ± 0.13 (*n* = 311)	20.76 ± 0.23 (*n* = 55)	20.86 ± 0.15 (*n* = 125)	20.70 ± 0.14 (*n* = 207)	20.90 ± 0.39 (*n* = 15)	20.94 ± 0.18 (*n* = 85)	20.69 ± 0.17 (*n* = 167)

Note: Values with different superscripts within the same line differ significantly at the *p* < 0.05 (a, b, c, d, e) level.

**Table 8 animals-13-02746-t008:** Typical frequencies of copy number variations within the *AKAP13* gene in goat.

Loci	Size	Genotypic Frequencies		
		Gain	Medium	Loss
CNV1	79	0.886 (*n* = 70)	0.079 (*n* = 6)	0.038 (*n* = 3)
CNV2	78	0.692 (*n* = 54)	0.090 (*n* = 7)	0.218 (*n* = 17)
CNV3	79	1.000 (*n* = 79)	-	-

**Table 9 animals-13-02746-t009:** Relationship between three InDel loci of the *AKAP13* gene and growth traits in SBWC goats.

Locus	Body Measurement Traits	Genotype (Mean ± SE)			*p*-Values
		II	ID	DD	
P1-16 bp InDel	Body height	55.29 ± 0.25 (*n* = 322)	56.64 ± 0.44 (*n* = 67)	56.50 ± 1.06 (*n* = 14)	0.827
	Body length	65.02 ± 0.39 ^a^ (*n* = 324)	67.02 ± 0.65 ^b^ (*n* = 67)	69.11 ± 1.61 ^b^ (*n* = 14)	0.010 *
	Height at hip cross	59.86 ± 0.34 (*n* = 159)	60.22 ± 0.51 (*n* = 51)	60.50 ± 1.04 (*n* = 10)	0.794
	Chest circumference	90.11 ± 2.09 (*n* = 324)	90.47 ± 0.99 (*n* = 67)	89.43 ± 2.12 (*n* = 14)	0.994
	Chest depth	27.02 ± 0.16 (*n* = 324)	26.71 ± 0.48 (*n* = 67)	28.11 ± 0.60 (*n* = 14)	0.286
	Chest width	17.45 ± 0.14 (*n* = 324)	18.03 ± 0.25 (*n* = 67)	18.04 ± 0.44 (*n* = 14)	0.177
	Cannon circumference	8.02 ± 0.05 (*n* = 324)	8.27 ± 0.10 (*n* = 67)	8.32 ± 0.17 (*n* = 14)	0.094
	Hip width	20.72 ± 0.18 (*n* = 154)	20.75 ± 0.24 (*n* = 50)	20.95 ± 0.44 (*n* = 10)	0.947
P2-15 bp InDel	Body height	56.95 ± 1.47 (*n* = 10)	57.03 ± 0.33 (*n* = 113)	56.17 ± 0.26 (*n* = 273)	0.169
	Body length	67.00 ± 3.09 ^AB^ (*n* = 8)	68.38 ± 0.56 ^A^ (*n* = 91)	66.23 ± 0.39 ^B^ (*n* = 250)	0.014 *
	Height at hip cross	62.60 ± 1.81 (*n* = 5)	60.75 ± 0.39 (*n* = 76)	59.83 ± 0.31 (*n* = 160)	0.080
	Chest circumference	89.20 ± 3.08 (*n* = 10)	95.88 ± 5.69 (*n* = 114)	90.53 ± 0.52 (*n* = 272)	0.345
	Chest depth	29.14 ± 0.67 (*n* = 7)	27.65 ± 0.32 (*n* = 103)	27.31 ± 0.18 (*n* = 213)	0.175
	Chest width	18.86 ± 0.86 (*n* = 7)	17.97 ± 0.25 (*n* = 103)	17.69 ± 0.17 (*n* = 214)	0.337
	Cannon circumference	7.96 ± 0.28 (*n* = 10)	8.23 ± 0.08 (*n* = 114)	8.20 ± 0.05 (*n* = 272)	0.600
	Hip width	20.80 ± 0.86 (*n* = 5)	20.94 ± 0.19 (*n* = 75)	20.68 ± 0.17 (*n* = 157)	0.651
P3-25 bp InDel	Body height	57.71 ± 0.75 (*n* = 21)	56.45 ± 0.34 (*n* = 121)	57.13 ± 0.39 (*n* = 93)	0.218
	Body length	70.29 ± 0.65 (*n* = 21)	69.31 ± 0.39 (*n* = 121)	70.39 ± 0.42 (*n* = 92)	0.146
	Height at hip cross	61.33 ± 0.58 (*n* = 21)	59.78 ± 0.36 (*n* = 121)	60.78 ± 0.38 (*n* = 93)	0.062
	Chest circumference	94.03 ± 1.17 (*n* = 21)	92.12 ± 0.65 (*n* = 121)	92.12 ± 0.61 (*n* = 93)	0.440
	Chest depth	29.05 ± 0.35 (*n* = 21)	28.10 ± 0.21 (*n* = 120)	28.09 ± 0.20 (*n* = 93)	0.146
	Chest width	17.90 ± 0.42 (*n* = 21)	18.29 ± 0.21 (*n* = 121)	18.24 ± 0.20 (*n* = 93)	0.746
	Cannon circumference	8.70 ± 0.13 (*n* = 21)	8.55 ± 0.05 (*n* = 121)	8.63 ± 0.05 (*n* = 93)	0.382
	Hip width	21.69 ± 0.38 (*n* = 21)	20.82 ± 0.20 (*n* = 117)	20.67 ± 0.18 (*n* = 92)	0.100

Note: All data was presented as mean ± SE. Values with different letters (a, b, A, B) differed significantly (*p* < 0.05). “*” means *p* < 0.05.

**Table 10 animals-13-02746-t010:** Statistical association analysis of three CNVs of AKAP13 with growth traits in goats.

Mutations	Growth Traits	Genotype (Mean ± SE)			*p*-Values
		Gain	Medium	Loss	
CNV1	Body height	60.00 ^a^ ± 0.44 (*n* = 70)	56.83 ^b^ ± 1.28 (*n* = 6)	56.67 ^ab^ ± 0.33 (*n* = 3)	0.045
	Body length	64.86 ^a^ ± 0.51 (*n* = 70)	60.58 ^b^ ± 1.61 (*n* = 6)	62.17 ^ab^ ± 3.20 (*n* = 3)	0.046
	Height at hip cross	61.80 ± 0.43 (*n* = 69)	60.83 ± 0.94 (*n* = 6)	59.83 ± 1.17 (*n* = 3)	0.532
	Chest width	14.81 ± 0.22 (*n* = 70)	14.50 ± 0.67 (*n* = 6)	14.00 ± 1.04 (*n* = 3)	0.701
	Chest depth	27.14 ± 0.28 (*n* = 69)	26.25 ± 0.40 (*n* = 6)	25.67 ± 0.33 (*n* = 3)	0.374
	Chest circumference	78.27 ± 0.72 (*n* = 70)	75.88 ± 0.97 (*n* = 6)	74.77 ± 1.63 (*n* = 3)	0.390
	Cannon circumference	7.52 ± 0.07 (*n* = 70)	7.25 ± 0.01 (*n* = 6)	7.43 ± 0.24 (*n* = 3)	0.493
CNV2	Body height	59.48 ± 0.51 (*n* = 54)	58.43 ± 1.19 (*n* = 7)	60.64 ± 0.89 (*n* = 17)	0.354
	Body length	64.76 ± 0.62 (*n* = 54)	62.64 ± 1.66 (*n* = 7)	64.41 ± 0.92 (*n* = 17)	0.493
	Height at hip cross	62.17 ± 0.51 (*n* = 53)	60.21 ± 1.08 (*n* = 7)	60.79 ± 0.64 (*n* = 17)	0.186
	Chest width	15.04 ± 0.23 (*n* = 54)	15.07 ± 0.72 (*n* = 7)	14.00 ± 0.42 (*n* = 17)	0.092
	Chest depth	27.57 ^a^ ± 0.29 (*n* = 53)	26.36 ^ab^ ± 0.52 (*n* = 7)	25.82 ^b^ ± 0.58 (*n* = 17)	0.011
	Chest circumference	79.24 ^a^ ± 0.77 (*n* = 54)	75.57 ^ab^ ± 0.95 (*n* = 7)	75.52 ^b^ ± 1.36 (*n* = 17)	0.026
	Cannon circumference	7.56 ^a^ ± 0.05 (*n* = 54)	7.79 ^ab^ ± 0.46 (*n* = 7)	7.20 ^b^ ± 0.11 (*n* = 17)	0.014

Note: Values with different superscripts within the same line differ significantly at the *p* < 0.05 (a, b) level.

## Data Availability

Data sets are available upon request by contacting the corresponding author.

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
