# Peer review of "InDel and CNV within the AKAP13 Gene Revealing Strong Associations with Growth Traits in Goat"

_animals, 2023, doi:10.3390/ani13172746_

Round 1

Reviewer 1 Report

The manuscript reports the results of a study aimed at analysing the relationships between polymorphisms in the AKAP13 gene and some growth traits in goats.
The study has positive aspects, but others need to be improved. My comments below.

Comments

L 40: ‘breed degradation’ is not a common definition. Please clarify what it mean

L 41: cultivating --> it is more appropriate for plants

L 48-49: as it is, the sentence is incorrect. Please rephrase

L 68-70: in which species?

L 86: more information on the goats sampled are necessary, such as sex and age, because they affect the measured traits

L 90-92: repetition

L 99-101: to build a DNA pool for subsequent detection of typing frequency --> not clear

L 101: I couldn’t find Liu et al. 2019 in the reference list

Table 3: chromas ?

L 130-131: instead of calculating Ho (homozygosity) and He (heterozygosity), which are complement to each other, it would be better to calculate the observed and the expected heterozygosity

L 137: over --> overall

L 147: were identified --> from M&M it seems that already identified polymorphisms were analysed

L 175-179: do not report the frequencies, they can be seen in the table

L 188: were  -->  are

L 191: actually seven haplotypes were detected, as haplotype 5 (DP1DP2IP3) has freq. = 0

L 194-195: if the loci are not linked, why to analyse the effect of the combined genotypes? Apart from that, the effects of the single loci are reported in Table 9, where the same general information can be found

L 223 and 272: ‘dominant’ can be ambigous, better to use ‘more frequent’

L 259-262: this is a questionable statement. The favourable effect of DP1DP1IP2DP2 is clearly due to DP1DP1 (tab. 9). Moreover, it is incorrect to compare DP1DP1IP2DP2 with DP2DP2

L315-317: it seems quite an overstatement to apply these conclusions to CNVs, considering the low sample size of some genotypic classes (especially for CNV1)

The English language is quite good

Reviewer 2 Report

Dear Authors

The work is very interesting. But add references I proposed and material and methode need details for some part as I mentioned in the paper. The bonferroni test need used to correct alpha.

Best rgards

Round 2

Reviewer 1 Report

This revised version has been improved and responds to most of my comments. Therefore, I only have a few additional comments.

Lines 193-194: I wonder whether it would be better to rename Haplo 6, 7 and 8 as Haplo 5, 6 and 7, adding a sentence such as 'The haplotype DP1DP2IP3 was not found'.

Lastly, I'm not satisfied with the response 19. It is true that increasing the number of tested animals wouldn't increase the frequency of the rare genotypes, but the absolute frequency would increase, so the group size could be increased, especially for CVN1 Loss (only 3 animals). In my opinion, with such low numbers, suggesting these markers for MAS seems a little bit much.

As for the language, it should be carefully revised in order to avoid the remaining mistakes (i.e.: at lines 237 and 276 more frequency --> more frequent).
